# Potential Explanatory Models of the Female Preponderance in Very Late Onset Schizophrenia

**Samantha Johnstone** [1] , **Gil Angela Dela Cruz** [1,2] , **Todd A. Girard** [3] , **Tarek K. Rajji** [4,5] and **David J. Castle** [1,2,6,*]

1. Centre for Complex Interventions, Centre for Addiction and Mental Health, Toronto, ON M6J 1H4, Canada
2. Institute of Medical Sciences, University of Toronto, Toronto, ON M5S 1A1, Canada
3. Department of Psychology, Toronto Metropolitan University, Toronto, ON M5B 2K3, Canada
4. Adult Neurodevelopment and Geriatric Psychiatry Division, Centre for Addiction and Mental Health, Toronto, ON M6J 1H4, Canada
5. Toronto Dementia Research Alliance, University of Toronto, Toronto, ON M5S 1A4, Canada
6. Department of Psychiatry, University of Toronto, Toronto, ON M1C 1A4, Canada
* Correspondence: david.castle@camh.ca

**Abstract:** Epidemiological and clinical studies have uniformly reported an overrepresentation of females with very-late-onset schizophrenia-like psychotic disorder (VLOS), in stark contrast to the sex distribution of early-onset schizophrenia. Various explanatory models have been proposed to account for these sex differences, including (a) antidopaminergic effects of estrogen, (b) differential vulnerability to subtypes, (c) neurodegenerative differences between the sexes, and (d) and sex differences in age-related psychosocial and neurological risk factors; however, these models have not yet been critically evaluated for their validity. Keywords related to VLOS symptomatology, epidemiology, and sex/gender were entered into the PubMed, MEDLINE, and Google Scholar databases spanning all years. Through a narrative review of symptomatology and pathophysiology of VLOS, we examine the strengths and limitations of the proposed models. We present a comprehensive biopsychosocial perspective to integrate the above models with a focus on the role of neuroinflammation. There is significant room for further research into the mechanisms of VLOS that may help to explain the female preponderance; the effects of estrogen and menopause, neuroinflammation, and dopaminergic transmission; and their interaction with age-related and lifetime psychosocial stressors and underlying biological vulnerabilities.

**Keywords:** very late onset schizophrenia; estrogen; female; late life psychosis; sex differences; neuroinflammation



## 1. Introduction

Schizophrenia is a serious mental illness characterized by positive symptoms (including delusions and hallucinations); disorganization symptoms (including formal thought disorder); negative symptoms (including restriction of affect, anhedonia, and amotivation); as well as cognitive deficits, such as reduced processing speed [1–3]. Typically, onset of schizophrenia occurs between the teens and late 20s ('early onset'), with antecedent prodromal symptoms often appearing during adolescence [4]. In approximately 15% of cases, however, schizophrenia first manifests after the age of 40 ('late onset'), with approximately 4% of cases beginning after the age of 60 (very-late-onset (VLOS); [5]).

In contrast to early onset schizophrenia, where incidence rates are predominantly 2–3 times higher in males [6], individuals diagnosed with VLOS are predominantly female [5]. This finding has been consistent across countries, methodologies, and diagnostic criteria employed (see Table 1), indicating that it is valid and not explicable merely in terms of the relative longevity of women [7]. Further consideration of the etiology of VLOS

and why it is a female preponderant disorder is necessary to direct future research and potentially find effective preventative measures.

**Table 1.** Published Studies of Sex Ratio in Late Onset and Very Late Onset Schizophrenia.

| Reference | Country/Setting | Diagnostic Criteria | Age of Onset | Number of Cases | F:M Ratio |
|---|---|---|---|---|---|
| Kay, 1963 [8] | County of Northumberland, UK | Late Paraphrenia | >65 | N = 57 | 5.3:1 |
| Herbert & Jacobson, 1967 [9] | UK | Late Paraphrenia (systematized delusions and/or hallucinations) | >65 | N = 47 | 22.5:1 |
| Bland, 1977 [10] | Canada | ICD-8 Schizophrenia (first admission) | ≥60 | N = 192 | 2:1 |
| Blessed & Wilson, 1982 [11] | Newcastle upon Tyne, UK | Late Paraphrenia[1] | ≥65 | N = 28 | 6:1 |
| Grahame, 1984 [12] | UK | Late Paraphrenia | ≥60 | N = 25 | 3.2:1 |
| Jørgensen & Munk-Jørgensen, 1985 [13] | Aarhus, Denmark | ICD-8 Schizophrenia and related disorders | ≥60 | N = 106 | 2.2:1 |
| Holden, 1987 [14] | UK | Late paraphrenia ('functional') | >60 | N = 34 | 7:1 |
| | | Late paraphrenia ('organic') | | N = 13 | 3:1 |
| Naguib & Levy, 1987 [15] | UK | Paraphrenia | ≥60 | N = 43 | 6.2:1 |
| Castle et al., 1993 [16] | UK | ICD-9 Schizophrenia and related disorders | >60 | N = 513 | 4.4:1 |
| Mazeh et al., 2005 [17] | Israel | DSM-IV Schizophrenia | ≥70 | N = 21 | 2.5:1 |
| Moore et al., 2006 [18] | South London and North West England | Very-late-onset Schizophrenia-like Psychosis (Howard et al., 2000) | >60 | N = 29 | 1.9:1 |
| Harris et al., 2014 [19] | Monash, Australia | DSM-IV-TR Delusional Disorder | >65 | N = 19 | 3.75:1 |
| Hanssen et al., 2015 [20] | The Netherlands | DSM-IV diagnosis of non-affective psychotic disorder | ≥60 | N = 28 | 7:1 |

Various models have been proposed to try to explain the differences in sex distribution of schizophrenia across the lifespan [5,6], including the following: (a) during reproductive age, females are "protected" from schizophrenia onset due to the antidopaminergic effects of estrogen; (b) females are differentially prone to certain subtypes of schizophrenia-spectrum disorder, which are more likely to manifest in later life; (c) VLOS is reflective of a subtype of neurodegenerative disorder; and (d) older females with an underlying biological vulnerability may be more likely than their male counterparts to experience psychosocial and neurological aging-related factors that are a pathway to VLOS. Notably, these models have not been evaluated for their validity and alone are limited in their ability to explain the female preponderance in VLOS, as is discussed below. Relevant to the present review, recent research has proposed a major role for neuroinflammation in the etiology of schizophrenia, with distinct sex- and age-related effects. Neuroinflammation in the central nervous system (CNS), in part caused by over-activation of microglia, represents a key factor responsible for onset and relapse of schizophrenia [21]. Microglia are brain macrophages responsible for immunity of the CNS [21] and are involved in synthesizing and releasing inflammatory

molecules (e.g., cytokines) in response to potential "intruders" and alerting other immune cells, clearing debris, and providing nutrients to repair damage [22]. When the exogenous threat has been resolved, the inflammatory response of microglia subsides and they return to a state of surveillance. Inappropriate activation of microglia has been implicated in numerous neurodegenerative and psychiatric disorders, including schizophrenia [21,23,24].

Sex and gender are complex constructs that need to be defined before further discussion regarding their role in onset of schizophrenia. As discussed by Sanchis-Sagura and Becker [25], sex refers to a biological trait that influences the development of the brain and body whereas gender refers to individual embodiments or expressions that interact with but may not align with, sex. Gendered environmental experiences, such as gender-based trauma or discrimination, and enforcement of gender roles, may subsequently influence the brain and alter behaviour. Thus, there is a bidirectional interaction between an individual's experience of their biological sex, their identified gender, and how the environment interacts with each of these domains; this in turn impacts clinically relevant distinctions in prenatal development, gonadal hormones, reactions to stressors, epigenetic constructions, neurodevelopment, and lived experiences [26]. Although the levels of circulating estrogen and other gonadal hormones decrease in females after menopause to the levels of age-matched males [27], sexually dimorphic effects endure and contribute to sex differences in aging related diseases [28–30], potentially including VLOS [31]. Throughout this paper, we refer to individuals based on their sex but highlight that any differences likely arise from an interaction of biological sex and gendered experiences.

Accordingly, this narrative review puts forward a comprehensive biopsychosocial perspective that attempts to integrate the prevailing models of sex differences in VLOS schizophrenia (see above), integrating emerging evidence on neuroinflammation in schizophrenia.

## 2. Results and Discussion

### 2.1. Epidemiology and Risk Factors Relevant to Sex Differences and Age at Onset of Schizophrenia

Numerous studies attest to the fact that females are more likely than males to be diagnosed with schizophrenia for the first time in later life (59). In individuals under the age of 40, there is a preponderance of males (around two thirds), whereas over the age of 60, the opposite occurs, with a females constituting 80–90% of reported cases [6,32,33].

Most studies on sex differences in schizophrenia focus on the male preponderance in those with early onset. Risk factors for early onset schizophrenia include viral infections during pregnancy, birth complications (e.g., preeclampsia and intra-uterine bleeding), substance use during adolescence (particularly cannabis use), stressful and/or traumatic lifetime experiences, genetic risk factors (i.e., polymorphic variations in brain-derived neurotrophic factor), amongst others [3]. Males appear to be more vulnerable to biological risk factors for schizophrenia, potentially due to slower prenatal maturation of the cerebral cortex and excessive synaptic pruning in adolescence [26]. Similarly, males diagnosed with schizophrenia also tend to have less social support and greater family discordance than females diagnosed with schizophrenia, which has negative impacts on treatment prognosis [26]. In addition, studies suggest that males with schizophrenia tend to have larger cerebral ventricles relative to healthy males, a finding not generally observed in females with schizophrenia relative to healthy females [34]. Some studies—although not all converge—have shown correlates of increased ventricular size with greater cognitive impairments, poor premorbid adjustment, more negative symptoms, clinical impairment, poor occupational record, and poor treatment response to antipsychotics [35].

The presentation of schizophrenia in females tends to be more atypical, with more affective features, fewer negative symptoms, fewer developmental and learning problems, better social adjustment, better treatment response, and lower rates of relapse relative to males [29]. Affective components present in females with schizophrenia may be impacted by gendered differences in socialization and trauma [26,35]. In line with this, there is less premorbid impairment in females with VLOS compared to males with early onset disorder [29,32]. Furthermore, females with early onset schizophrenia and individuals

with VLOS are both more likely to be married, to be less socially compromised, and to have better premorbid vocational and work histories, relative to males with early onset schizophrenia [6,32]. Better social skills and neuropsychological functioning in females with schizophrenia may in part reflect better interpersonal relationships, lower rates of disease related stigma, and greater tolerance by patients for experience of symptoms [36].

Neill et al. [37], Chen et al. [38], and Selvendra et al. [39] examined the relationships between age of onset of schizophrenia and sex using data from the Australian Survey of High Impact Psychosis (SHIP). The SHIP study had an age cut-off at 65 years so cannot address VLOS, but findings regarding sex differences remain instructive to the present discourse. Notably, SHIP found that 12.7% of the variance in age of onset in women could be attributed to risk factors including immigration status, loss of a family member, poor premorbid social adjustment, and family history of psychiatric illness. Specifically, earlier age of onset in females was associated with the loss of a close relative, substance use, a familial history of psychosis, poor premorbid work and social adjustment, and the experience of a premorbid personality disorder as compared to females with later onset (see Table 2). Physical abuse and childhood sexual abuse in females were not exclusive to early onset schizophrenia, but were associated with earlier age of onset. Other studies have similarly found that in VLOS, family history of schizophrenia and early childhood maladjustment are not frequent as in early onset schizophrenia [21,40]. Of interest, earlier literature in this area implicated uncorrected sensory deficits (e.g., deafness but no hearing aid) as a risk factor for late life development of schizophrenia and other psychoses; however, causality has not been determined and might reflect impaired help-seeking behaviours [41].

**Table 2.** Risk Factors for Very Late Onset Schizophrenia.

| Risk Factor | Early Onset | Very Late Onset Schizophrenia | Reference |
|---|---|---|---|
| Sex | Male | Female | Castle et al., 1993 [16] |
| Pregnancy and birth-related complications | More prevalent | Less prevalent | Janoutová et al., 2016 [3] |
| Traumatic life experiences | More prevalent | Less prevalent | Janoutová et al., 2016 [3] |
| Genetic risk factors | More prevalent | Less prevalent | Janoutová et al., 2016 [3] |
| Ventricle size | Larger | Not observed | Castle & Murray, 1991 [35] |
| Poor premorbid social adjustment | More prevalent | Less prevalent | Neill et al., 2020 [36] |
| Family history of psychiatric illness | More prevalent | Less prevalent | Neill et al., 2020 [36] |
| Family history of schizophrenia | More prevalent | Less prevalent | Ayano, 2016 [39] |
| Loss of close relative | More prevalent | Less prevalent | Neill et al., 2020 [36] |
| Alcohol or drug abuse | More prevalent | Less prevalent | Selvendra et al., 2022 [38] |
| Immigration status | Non-migrant status | Migrant status | Neill et al., 2020 [36] |
| Physical and childhood sexual abuse | More prevalent | Less prevalent | Selvendra et al., 2022 [38] |

*2.2. Symptomatology and Pathophysiology of Onset Categories*

Relevant to the present review, differences in symptomatology of early onset and VLOS may reflect differences in etiological mechanisms and/or neurobiological sex differences interacting with the risk factors outlined above and in Table 1 [35]. With respect to early onset schizophrenia, patients usually experience both positive, disorganisation, and negative symptoms, along with a range of cognitive deficits in nearly all domains (i.e., attention, executive functioning, processing speed, social cognition, memory, working memory, etc.), with a range of severity [1,40].

Comparatively, individuals with VLOS rarely manifest disorganization symptoms or primary negative symptoms (Table 3). Conversely, positive symptoms are often floridly present, with particularly high rates of paranoid and persecutory delusions, partition delusions, and hallucinations in a number of modalities (see Table 3; [20,42–44]). Relative to healthy adults, individuals with VLOS also experience higher rates of social isolation, poor physical health, lower education and occupational opportunities, and depressive symptoms across the lifespan [38,45], potentially indicative of some prodromal symptomatology that

did not reach threshold. However, lifetime social functioning is on average better than early onset counterparts, with marriage being more common [6,32].

**Table 3.** Differences in Presentation of Schizophrenia Based on Age of Onset.

| Features | Earlier Onset | Later Onset | Reference |
|---|---|---|---|
| Positive Symptoms | Less Prevalent | More Prevalent | Hanssen et al., 2015 [20] |
| Partition Delusions | Less Prevalent | More Prevalent | Van Assche et al., 2017 [44] |
| Paranoid Delusions | Less Prevalent | More Prevalent | Van Assche et al., 2017 [44] |
| Persecutory Delusions | Less Prevalent | More Prevalent | Castle et al., 1997 [6] |
| Hallucinations | More Prevalent | Less Prevalent | Castle et al., 1997 [6] |
| Psychotic Episodes | More Prevalent | Less Prevalent | Hanssen et al., 2015 [20] |
| Negative Symptoms | More Prevalent | Less Prevalent | Howard et al., 2000 [5] |
| Restricted affect | More Prevalent | Less Prevalent | Castle et al., 1997 [6] |
| Formal thought disorder | More Prevalent | Less Prevalent | Almeida et al., 1995 [46] |
| Inappropriate affect | More Prevalent | Less Prevalent | Castle et al., 1997 [6] |
| Lifetime diagnosis of Major Depression | More Prevalent | Less Prevalent | Hanssen et al., 2015 [20] |
| Cognitive deficits | Significant deficits | Some deficits | Van Assche, et al., 2017 [44] |
| Intelligence | Significant deficits | Some deficits | Vahia et al., 2010 [47]; Sachdev et al., 1999 [48] |
| Processing Speed | Significant deficits | Some deficits | Vahia et al., 2010 [47]; Sachdev et al., 1999 [48] |
| Executive Functioning | Significant deficits | Significant deficits | Brichant-Petitjean et al., 2013 [49]; Hanssen et al., 2015 [20] |
| Attention | Significant deficits | Significant deficits | Brichant-Petitjean et al., 2013 [49]; Hanssen et al., 2015 [20] |
| Verbal learning and memory ** | Some deficits | Some deficits | Hanssen et al., 2015 [20] |
| Social cognitive functioning | Primarily impaired | Primarily intact | Moore et al., 2006 [18]; Smeets-Janssen et al., 2013 [50] |

** Note: Mixed findings with respect to findings of deficits in verbal learning and memory; see [44]

Furthermore, individuals with VLOS experience significant impairments in general cognitive functioning, beyond what is expected in nonpathological aging but comparable to dysfunction present in older adults with early onset schizophrenia [44]. Specifically, similar impairments are observed across onset categories in intelligence, attention, processing speed, and executive functioning, which are distinct from neurodegenerative conditions [20,44]. Having said this, cognition tends to be more intact in VLOS compared to aging individuals with early onset schizophrenia; meta-analyses have suggested a compensatory mechanism for cognitive functioning in later onset cases [51]. However, language deficits in VLOS more closely resemble Alzheimer's disease [44]. Interestingly, people with VLOS tend not to have significant impairments in domains of social-cognitive functioning, despite presence of paranoia and this being a predominant impairment in early onset schizophrenia [44]. This may be related to greater levels of social participation [6,32] that occurred over the lifetime prior to onset.

In line with similarities in neurocognition amongst onset categories, a large systematic review found that neurobiological underpinnings of neurocognitive deficits are similarly comparable [44]. Specifically, as in early onset patients, larger cerebral ventricles occur in VLOS, distinct from what is observed in Alzheimer's disease [44]. Other studies have reported similarities across age-at-onset groups in terms of lower volumes of the amygdala, entorhinal cortex, and left hippocampus across onset categories, as well as smaller anterior and superior temporal lobes and white matter pathology, all of which is distinct from

what is observed in neurodegenerative diseases [44]. Furthermore, post-mortem studies show no evidence of amyloid deposits as is characteristic of Alzheimer's disease, although some cases show evidence of Lewy body pathology and corticobasal degeneration [44]. Thus, while there is some disagreement as to whether VLOS is a neurodegenerative condition or whether differences are reflective of aging combined with schizophrenia-related deficits [17,21], recent research appears to support the latter [44]. It should be noted that there is likely heterogeneity as some subgroup do experience neurodegeneration (one study found 35% prevalence of dementia in 37 patients in a follow up of 10 years or until death [14]); however, it is not clear whether this is due to VLOS, misdiagnosis, or a co-morbid condition [14,52]. Notably, a large study on 8062 individuals found that late onset psychosis is a risk factor for the development of dementia [53], suggesting comorbidity.

### 2.3. Sex Differences in Other Late-Life-Onset Disorders with Potential Psychosis

Before considering competing explanatory models for the female preponderance in VLOS, it is instructive to appraise the literature on sex differences in other late-life-onset disorders in which psychotic symptoms might manifest.

### 2.3.1. Alzheimer's Dementia

Alzheimer's disease (AD) is a neurodegenerative disorder associated with hyperphosphorylated tau tangles and β-amyloid plaque deposition; it is the leading cause of dementia worldwide. Neuroinflammation—and particularly microglial overactivation—plays a key role in the pathogenesis of AD [54]. Specifically, reactive microglia have been shown to localize in proximity to amyloid plaques, resulting in overactivation of microglia, which leads to phosphorylation of tau [54]. Psychotic-like symptoms, particularly persecutory beliefs, are common in AD (~41% of cases) [55]. Similar to VLOS, there is a preponderance of the diagnosis in females (2:1; [56]). Moreover, females with AD experience faster cognitive decline than men with AD [57]. In terms of symptoms, a recent review reported that females with AD were more likely than males with AD to have severe depressive symptoms, delusions, and aberrant motor behaviour, whereas there were no sex differences in other psychiatric symptoms [58]. In trying to explain these sex effects in AD, estrogen has been suggested to be a neuroprotective factor, and treatment with estrogen replacement therapy post-menopause has been shown to support cognitive function and to reduce risk of disease onset in females [59]. The rapid depletion of circulating estrogen during and after menopause appears to contribute to the increased risk of AD onset in females at that life-stage [60].

### 2.3.2. Depression

A female preponderance in depression pertains across the lifespan, including in late life, with females being more likely than males to experience a first episode between the ages of 70 and 85 [61]. Of relevance to the current discourse, rates of psychotic depression also tend to increase with age, both in those with early onset depression and late onset depression [62]. Psychotic depression appears to be more prevalent in older-adult females relative to males [63,64], albeit studies are sparse and have small sample sizes. Neuroinflammation, particularly overactivation of microglia, has been implicated in geriatric depression, via effects on cognitive and emotional networks in the CNS [65]. Indeed, high levels of peripheral cytokines and elevated C-reactive protein precede onset of geriatric depression [65].

### 2.3.3. Parkinson's Disease

Parkinson's disease (PD) is a progressive neurodegenerative disorder with motor symptoms (e.g., tremor, muscle rigidity) as well as cognitive and psychiatric symptoms (e.g., psychosis, depression) [66]. Females with PD tend to show greater rates of depressive symptoms but similar rates of psychotic symptoms relative to males with PD [67]. PD is primarily thought to result from cellular death of dopaminergic neurons in the basal

ganglia [66]. Distinct from other disorders discussed thus far, PD is significantly more likely to occur in males and when it does occur in females, onset is usually later in life (53.4 years old versus 51.3; [68]). Interestingly, age of onset in females has been positively correlated with later menopause and greater fertile lifespan, compatible with a 'protective' effect of cumulative estrogen levels on risk [68,69]. However, disease progression after onset is reportedly accelerated in females [70]. Furthermore, dopaminergic cells show less susceptibility to degeneration in females relative to males [22]. Chronic neuroinflammation has also been implicated in the pathogenesis of PD, such that haploinsufficiency of protective midbrain factors resultant from environmental triggers (e.g., aging and chronic illness) may increase vulnerability to inflammation-induced dopaminergic neuronal death [71].

### 2.3.4. Delusional Disorder

Delusional disorder is a serious psychiatric disorder that usually onsets in middle-late life and includes a range of delusional symptoms, although hallucinations are not usually present (and if they are, are not a prominent part of the mental state). Further, disorganization symptoms and cognitive deficits are not prominent, as distinct from schizophrenia [72]. However, overlap of delusional beliefs may suggest similarities in risk factors and vulnerabilities to VLOS. Epidemiological and clinical data on delusional disorder are limited. There are no major sex differences in incidence and prevalence, albeit there is reportedly higher incidence in females post-menopause relative to pre-menopause [73]. Potential risk factors include genetic vulnerability, immigration, minority status, socioeconomic status, medical comorbidities, and sensory impairment, as well as potentially estrogen loss [73]. Explanatory models of onset of delusional disorder and sex differences have not yet been thoroughly investigated, but the finding that females are more prone in the post-menopausal stage of life is informative and relevant to the current paper.

### 2.4. Explanatory Models of Sex Differences in Very Late Onset Schizophrenia

As detailed above, sex difference in the prevalence of VLOS is consistent across studies, including both population and clinical sampling [6,32,33]. A number of models (Figure 1) have been proposed to explain the sex difference in vulnerability to onset of schizophrenia and related psychoses at different ages. In summary, these models propose the following:

(a) During reproductive age, females are "protected" from schizophrenia onset due to antidopaminergic effects of estrogen, wherein estrogen regulates dopaminergic transmission, "sparing" females from early onset of schizophrenia [74]. Studies have found that estradiol may increase the threshold for psychosis through regulation of dopaminergic transmission and degradation, particularly in the mesolimbic and mesocortical pathways, and in reducing oxidative stress [31]. Notably, females with early onset schizophrenia are more likely to be hypoestrogenic (i.e., lower circulating levels of estrogen), evidenced by menstrual irregularities [75–77]. Low estrogen is suspected to precede onset of schizophrenia due to later menarche, infertility, and lighter menstrual periods prior to symptom emergence [78]. However, effects of estrogen on dopamine alone cannot explain the observed increase in diagnoses of females with VLOS, as onset is several years after menopause has occurred (typically age 50; [79]) and circulating estrogen levels have been low and approximately equal to age-matched males for several years [27].

(b) VLOS may reflect a subtype (or subtypes) of schizophrenia that is (are) more likely to onset later in life, to which females are more vulnerable [35]. Stringent diagnostic criteria for schizophrenia, relative to other schizophrenia-spectrum disorders, may exclude more females, as they tend to present more 'atypical' features, including marked affective symptoms in some cases. Diagnostic criteria for schizophrenia that emphasize negative symptoms and psychosocial decline may exclude more females than males across the lifespan, especially in later life [16,35]. Differential vulnerability to certain presentations or subtypes (e.g., VLOS) based on sex may explain the female preponderance as well as variations in symptom and presentation between onset

categories. However, this explanation serves as more of a typological characterization, while mechanisms of development, presentation, and the cause of the sex distribution remain unanswered.

(c)  VLOS is reflective of a type of neurodegenerative disorder to which females are differentially prone. The neurodegenerative hypothesis may explain the predominance of positive symptoms and the absence of negative symptoms, as neurodegenerative conditions such as AD are often accompanied by hallucinations and paranoid delusions [56]. Disorders such as AD similarly have a female preponderance as well as etiological mechanisms that implicate neuroinflammation. However, as discussed above, research does not support progressive neurodegeneration in most cases of VLOS, suggesting that it is not primarily a neurodegenerative disease. There are substantial similarities in neurocognitive profiles between VLOS and other onset categories of schizophrenia as well as substantial differences relative to other neurodegenerative conditions. Furthermore, in VLOS there is a lack of neurobiological evidence of atrophy and tauopathy [44], hallmarks of neurodegeneration [80]. In fact, VLOS is a risk factor for the development of dementia and other neurodegenerative diseases [20,44,53].

(d)  Psychosocial aging-related factors may be more likely to occur in females (e.g., bereavement, job loss, etc.) in part due to their generally longer lifespan [7], and these may interact with neurological aging-related risk factors to contribute to onset of schizophrenia in late life [40]. However, specificity of psychosocial risk factors to onset of schizophrenia is unclear, as these can precipitate several late-life conditions, including AD [81] and geriatric depression [82]. Furthermore, the sex distribution does not appear to be entirely explained by this model, as there is no consistent evidence of a gender difference in negative life events related to aging, other than being married and subsequently widowed, which is significantly more prevalent in older females [61]. Comparatively, aging-related neurological changes may explain the very late life onset and sex distribution. For example, tardive dyskinesia is significantly more likely to occur in females over 70 years old, and symptoms are generally more severe than in males, supporting aging-related and sex-divergent dopaminergic dysfunction. Similarly, females experience significant age-associated decreases in striatal dopamine transporter that is not paralleled in males [83]; this may contribute to the dysregulation of dopaminergic signaling that is present in schizophrenia [74,84]. However, other studies have not supported the etiological role of the dopamine transporter in etiology of schizophrenia, limiting this model's ability to fully explain VLOS [85], and the mechanisms of sex-specific dopaminergic effects are still unclear.

In recent years, further research on the etiology of schizophrenia and symptomatology across the lifespan [23] suggest that a biopsychosocial mechanistic perspective is warranted and that neuroinflammation may be the 'missing link' across these explanatory hypotheses.

### 2.5. Towards a Unified Theoretical Approach

2.5.1. Neuroinflammation in Schizophrenia

We first overview the etiological role of neuroinflammation in schizophrenia before exploring how it may differ by sex and potentially explain differences in onset of VLOS. In individuals with schizophrenia, increases in inflammatory cytokines and related biomarkers are seen in the prodromal stage and during acute illness episodes, with inflammation decreasing after pharmacological interventions [86–88]. Furthermore, positron emission tomography (PET) studies have identified overactive microglia in individuals at high risk of developing schizophrenia and those with established schizophrenia; these findings are supported by post-mortem studies that have identified greater density of microglia in people with schizophrenia relative to controls [21]. Over-activation of microglia is implicated in inappropriate synaptic pruning that has been suggested as a key factor in the pathophysiology of schizophrenia [86–88]. Furthermore, shortened telomeres, indicative of

biological aging and cellular dysfunction [89], tend to be present in the microglia of people with schizophrenia [90,91]. While some studies have reported no microglial over-activation in schizophrenia, many of these did not control for use of antipsychotic drugs, which have an inhibitory effect on neuroinflammation [21]. Of particular relevance, individuals with VLOS have elevated blood concentrations of inflammatory markers (e.g., C-reactive protein; [92]), which is associated with the severity of cognitive impairments in aging populations [93]. Microglial-mediated neuroinflammation appears to explain disease progression such that animal models show microglial activation growing steadily over the lifespan, coinciding with emergence of behavioural impairments [24]. Furthermore, increased microglial activation is significantly related to greater positive symptom severity in people with schizophrenia [94] and individuals with high levels of persecutory ideation tend to show greater microglial activation [23].

| | |
|---|---|
| **A: Antidopaminergic effects of estrogen** | • At age 60, circulating estrogen has been low for several years<br>• Effects on dopamine neurons diminish quickly |
| **B: More vulnerable to subtypes that occur later** | • Typological characterization<br>• Mechanisms still unclear |
| **C: VLOS is a neurodegenerative disorder not schizophrenia** | • Substantial differences between VLOS and other neurodegenerative conditions<br>• More similarities between typical onset schizophrenia and VLOS |
| **D: Psychosocial aging-related stressors preceding onset more likely to occur in females** | • Specificity to schizophrenia onset not explained<br>• No consistent evidence of a gender difference in negative life events (other than being widowed) |

**Figure 1.** Limitations of the early potential explanatory models of the female preponderance in very late onset schizophrenia.

The prevailing hypothesis regarding the pathophysiology of schizophrenia attributes symptoms and cognitive deficits to hyperactive dopaminergic transmission in mesolimbic and striatal brain regions and hypoactive transmission in the prefrontal cortex, with cascading effects on γ-Aminobutyric acid (GABA), acetylcholine, and serotonin transmission [74]. Importantly, dopamine has been shown in animal studies to exert a regulatory effect on microglial cells [95,96], thereby reducing inflammatory responses. Conversely, low dopamine levels may selectively stimulate high affinity dopamine receptors in microglia, promoting inflammatory responses [97,98]. Neuroinflammation may in turn affect expression of dopamine receptors, altering signaling [97]. Thus, aberrant dopaminergic transmission present in schizophrenia may be a contributing factor to the observed dysregulation of inflammatory responses observed, and vice versa.

Within the general population, females tend to have higher inflammatory markers relative to their male counterparts [22] but are also less susceptible, pre-menopause, to

negative effects of neuroinflammation [99]. With respect to early onset schizophrenia, one study found that males had a greater presence of neuroinflammatory biomarkers (e.g., malondialdehyde (MDA)) relative to male controls, whereas this was not the case for female schizophrenia patients relative to female controls [100]. Furthermore, there was no significant difference between male and female schizophrenia patients with respect to MDA, whereas in healthy controls, females had significantly higher rates of MDA relative to males. Moreover, higher levels of MDA and tumor necrosis factor alpha (TNF-$\alpha$) correlated with more prominent psychotic symptomatology [100]. These findings have been supported by other studies of MDA in schizophrenia [101,102]. Thus, evidence regarding neuroinflammation in males with early onset schizophrenia may reflect abnormal inflammatory responses, whilst in females, other pertinent risk factors (e.g., susceptible genetics, high childhood adversity [40]) or dysregulated estrogen levels that are unable to counteract effects of inflammation [75–77], may play a greater role in development of the disorder.

### 2.5.2. Relevance of Estrogen to Very Late Onset Schizophrenia

Estrogen plays a key role in reducing neuroinflammation and in regulating dopamine synthesis, degradation, and reuptake, along with other protective cellular functions [22]. Estrogen receptors are expressed in all neural cells and activation of receptors in several cell systems may be neuroprotective via modulation of microglial activation [103]. For example, animal studies have shown that administration of estradiol during proinflammatory responses blocks activation of microglia [104]. Mouse models further demonstrate that low levels of circulating estrogens are related to greater susceptibility of aged microglial cells to produce and maintain inflammatory responses [105]. Also of relevance is that females tend to have higher blood flow in the brain from mid-adolescence to approximately the age of 60 (aligning with age of onset of VLOS), and this may play a role in differential activation of microglial cells or sex-specific microglial subpopulations in CNS regions [106]. In humans, it has been observed that, after menopause, females exhibit a significant increase in the expression of inflammatory biomarkers [22]. In both males and females, aging is associated with increased inflammation [107], and this effect is pronounced in females, particularly those aged 51–60 [107]. Around this age, menopause occurs and rapid hormone depletion induces profound changes in microglial activity along with increased evidence of neuroinflammation [106]. Thus, aging-related resistance coupled with lower circulating estrogen in older females may significantly impair the resolution of inflammatory responses, with cascading effects on dopaminergic signaling [97,98].

Appropriately functioning estrogens may be beneficial to females vulnerable to schizophrenia via modulation of neurological risk factors including oxidative stress, neuroinflammation, and dopamine transmission. However this effect diminishes post-menopause (7) with cellular estrogen withdrawal exacerbating effects [106]. Direct relevance of this effect to onset of schizophrenia in postmenopausal females has not yet been established and the link to VLOS is tenuous [52], albeit parsimonious. It is feasible that there are indirect and enduring effects such that gradually increasing neuroinflammation [22,31] may explain the delay from menopause to onset of VLOS. This is supported by the fact that biomarkers of neuroinflammation and dysregulated microglial cells tend to be greatest around age 60 [107]. Of note, this is distinct from model A described above, implicating direct antidopaminergic effects of estrogen that immediately diminishes post-menopause (Figure 1).

### 2.5.3. Psychosocial Aging-Related Factors, Neuroinflammation, and Telomeres in Schizophrenia

As mentioned above, neuroinflammation appears to explain disease *progression* in schizophrenia [24]. However, the relationship is somewhat more complex as evidence suggests that psychosocial-related stressors may in turn induce neuroinflammation [108]. For example, animal models of depression have found that chronic mild external stress

results in central and peripheral nervous system inflammation, indicated by high levels of inflammatory biomarkers (e.g., TNF-$\alpha$ and reduced brain-derived neurotrophic factor). Chronic psychosocial stress has also been implicated as a contributory risk factor (along with neurobiological vulnerabilities) to late-onset AD [109,110]. It has further been suggested that psychosocial stress impacts microglial activity, contributing to the pathological progression of AD [111] and potentially VLOS.

Telomeres are regions at the ends of chromosomes that shorten with each cell division, with lengths being indicative of an individual's biological age [89]. Individuals with schizophrenia have cells, particularly microglia, with telomeres shorter than expected for their biological age [90,91]. Meta-analyses suggest that experiencing symptoms of schizophrenia and the associated psychobiological stress precedes telomere shortening, potentially indicative of a causal effect [91]. Telomere length in people with or at risk for schizophrenia may further be affected by factors such as age; sex; smoking [112]; trauma [113]; social isolation and loneliness [114]; as well as inflammation, oxidative stress, and premature aging [115,116]. In healthy populations and in patients with schizophrenia over the age of 50, females generally have longer telomeres relative to their male counterparts [112,117], which may be explained—in part at least—by the neuroprotective effect of estrogen in younger age [118]. Further research into accelerated telomere shortening in individuals with VLOS may also indicate abnormal functioning of microglial cells, such that post-menopause, aged microglial cells produce greater inflammatory responses, which are no longer modulated by estrogen [105]. Similar mechanisms have been implicated in other geriatric onset disorders [119], including the brain-age gap hypothesis that suggests accelerated biological aging in numerous mental illnesses (e.g., schizophrenia, posttraumatic stress disorder, early and geriatric onset depression; [120]).

### 2.5.4. Summary of Model

Over adolescence and adulthood, some females who are vulnerable to schizophrenia may experience high life stress but do not develop the disorder (Figure 2, [6,38]). These stressors may be linked to prodromal symptoms, including social isolation, financial problems due to difficulty with employment, substance misuse, and trauma amongst others [38], and contribute to aging of microglial cells, which could be indicated by shortened telomeres [90,91,113]. After menopause (age 40–58; [79]), circulating estrogen levels rapidly decrease, and overactive microglia and dysregulated dopamine are no longer modulated by estrogen. The subsequent rapid decrease in estrogen affects cellular functioning and inflammatory responses, beyond what occurs in age matched males [121], further dysregulating dopaminergic signaling [97,98]. Recent studies confirm the exacerbating effects of menopause in females with schizophrenia, including worsening of symptoms and changes to treatment efficacy [122–124], supporting a role of hormonal changes in disease progression. Thus, coupled with aging-related psychosocial factors (e.g., bereavement and job loss) and neurological changes in the aging brain, menopause-related hormone changes may result in females who were vulnerable to schizophrenia transitioning to disorder onset in the ensuing years (a feature similar to delusional disorder), thus explaining the high preponderance of females diagnosed with VLOS [5,6]. Indeed, low levels of estrogen and high levels of neuroinflammation may explain certain diagnostic differences between onset subtypes, particularly high levels of positive and affective symptoms [6].

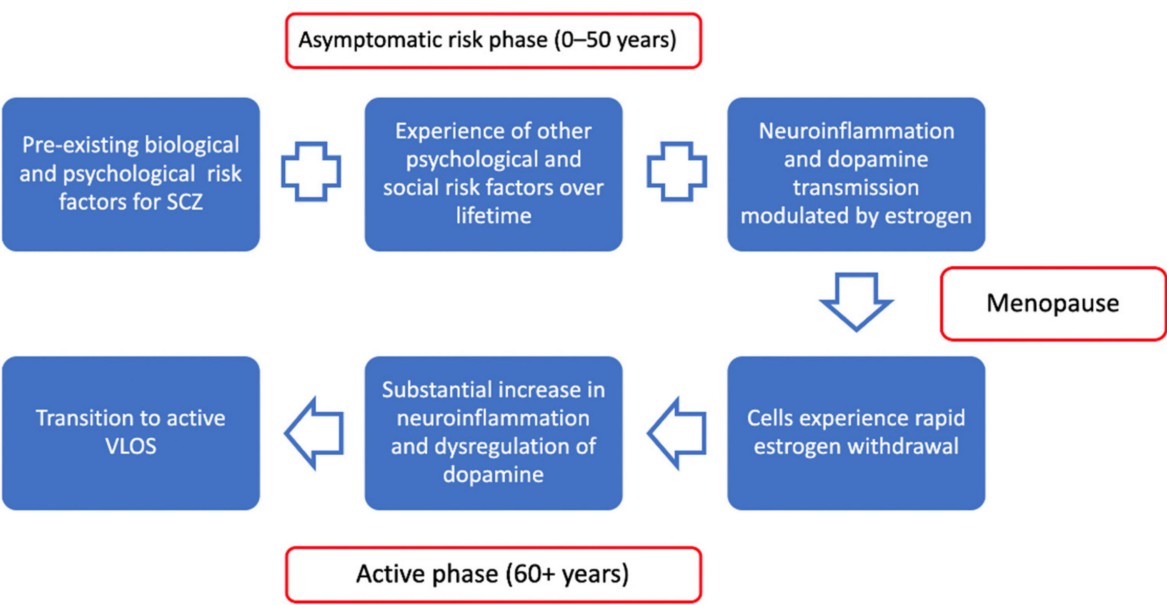

**Figure 2.** Figure depicts summary of proposed model for the female preponderance in very late onset schizophrenia.

### 3. Limitations

There is sparse research with respect to neuroinflammation in individuals with VLOS, and as such, we are unable to draw definitive conclusions. In addition, interactions between gonadal hormones and inflammatory reactions are highly complex and additional factors not discussed here may also be relevant. For example, testosterone has also been shown to have anti-inflammatory effects [125]. The relative contribution of testosterone and estrogen along with other sex hormones (e.g., progesterone), and how these may differ based on an individual's sex require further attention.

### 4. Materials and Methods

#### 4.1. Search Strategy and Inclusion Process

Although this is a narrative review, we utilize a systematic approach to searching for studies so as to incorporate all relevant literature into our paper and to formulate an unbiased model. Key words related to VLOS symptomatology, epidemiology, and sex/gender were entered into PubMed, MEDLINE, and Google Scholar databases spanning all years by SJ and GD (for all search terms see Supplementary Materials). Studies were included if they were found in English; published in peer-reviewed and respected journals; and investigated symptomatology, epidemiology, and sex/gender differences in VLOS. Comparison studies for early onset schizophrenia and other late-onset psychiatric and neurological disorders were also selectively searched for. Reference lists of identified papers were checked for additional relevant literature.

#### 4.2. Synthesizing Approach

The findings were synthesized using a narrative approach. We first provide an overview of similarities and distinctions amongst age-at-onset categories, with a focus on studies reporting sex differences in schizophrenia and VLOS specifically. Where appropriate, we make reference to sex differences in other late-onset brain diseases. Then, we summarize the strengths and limitations of models that have tried to account for sex difference in VLOS and discuss the potential role of neuroinflammation, dopamine, estrogen, and psychosocial risk factors in the development of VLOS. Tentative conclusions on the available evidence are drawn.

## 5. Conclusions

There is significant room for further research into the mechanisms of VLOS that may help to explain the female preponderance. Drawing from evidence in early onset schizophrenia as well as other late-onset psychiatric and neurological disorders suggests that the neuroprotective effect of estrogen and its subsequent loss post-menopause on microglial overactivation and dopaminergic transmission, interacting with age-related and lifetime psychosocial stressors and underlying biological vulnerabilities, likely play key roles in VLOS. This model incorporates a biopsychosocial perspective looking at changes in risk factors across the lifespan to explain the female preponderance in VLOS. Importantly, a clear understanding of the etiology of VLOS is crucial to the development of effective preventative and responsive treatments.

**Supplementary Materials:** The following are available online at https://www.mdpi.com/article/10.3390/women2040033/s1.

**Author Contributions:** S.J.: methodology, investigation, writing—original draft, visualization, G.A.D.C.: writing—original draft, writing—review and editing, methodology, T.A.G.: writing—review and editing, T.K.R.: writing—review and editing, D.J.C.: writing—review and editing, conceptualization, supervision. All authors have read and agreed to the published version of the manuscript.

**Funding:** This research received no external funding.

**Institutional Review Board Statement:** Not applicable.

**Informed Consent Statement:** Not applicable.

**Data Availability Statement:** Not applicable.

**Conflicts of Interest:** The authors declare no conflict of interest.

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
