# Peer review of "Potential Explanatory Models of the Female Preponderance in Very Late Onset Schizophrenia"

_women, doi:10.3390/women2040033_

Round 1

Reviewer 1 Report

This is a thorough review and well-reasoned assessment of possible explanations for the predominance of women in incident very late onset schizophrenia. The authors review the major hypothesized explanations and identify strengths and weaknesses of each. They propose a unifying mechanism of neuro-inflammation and identify limitations and strengths of this proposal.

My only concern is a minor one. The title suggests that it answers the question of why a female predominance when the article makes it clear that an answer is not clear. 

Author Response

Reviewer 1:

This is a thorough review and well-reasoned assessment of possible explanations for the predominance of women in incident very late onset schizophrenia. The authors review the major hypothesized explanations and identify strengths and weaknesses of each. They propose a unifying mechanism of neuro-inflammation and identify limitations and strengths of this proposal.

My only concern is a minor one. The title suggests that it answers the question of why a female predominance when the article makes it clear that an answer is not clear. 

Response: Thank you for this comment. We have changed the title to the following:

Potential Explanatory Models of the Female Predominance in Very Late Onset Schizophrenia

Reviewer 2 Report

This is a very interesting paper about very late onset schizophrenia an gender differences in the occurrence of this clinical condition. The paper is well-written, and is of interest for the readers. However, several minor changes should be made before considering it for publication.

Abstract.

1- I recommend to follow the usual abstract structure: Introduction, Aims, Methods, Results and Conclusions. Please, describe the main methods of this narrative review. How did the authors the search? What key words did they use to find the included studies? This should be briefly reported in the abstract.

2- Conclusion. I recommend to draw some conclusions in line with the main questions that the authors addressed.

3- "Later life" is correct instead of "later lafe". Please, correct this mistake.

Introduction

1- The definition of schizophrenia (and symptom domains) should be referenced with at least two or three recent studies. Perhaps, the authors should also consider to include the DSM-5 description (briefly reported).

2-The effect of menopause in the appearance of schizophrenia in later life should be described in the introduction section. At the end of the introduction, the authors report that sex and gender are complex constructs that need to be defined. Please, expand this section to justify why the authors draw the hypothesis about the sex differences at this age.

3- At the end of the introduction, I recommend to repeat those questions to be addressed that were reported in the abstract. "Various explanatory models (mentioned in the introduction) have been proposed: a) during reproductive age (...), b) females are differently prone (...), c) older females may be more likely (...).

4-The main aims of this narrative review should be reported at the end of the introduction. I recommend to report them in line with the questions to be addressed.

Methods

1- I recommend to divide the methods section into several subsections. Screening and selection processes; inclusion/exclusion criteria, key-words used, etc.

Results

1- The results section should be divided into several subsections. These should be numbered. 

2-Sex differences in Other Late-Life-Onset Disorders with Potential Psychosis. The authors described some words about Alzheimer's Dementia, Depression and Parkinson's Disease. Delusional Disorder should be included as it is a disorder frequently occurring at the middle- late life, and in several cases with very late onset.

3- The section about "Towards a Unified Theoretical Approach" is a kind of discussion of the overall results. I recommend to call it "Discussion" and maintain the same subsections ("Neuroinflammation, Relevance of Estrogen, Psychosocial Aging-Related Factors).

4- In the summary of the model section, the authors describe some words about "menopause". There are several recent works on this topic, summarizing the effect of menopause in the context of schizophrenia. I recommend to expand this section by discussing more in depth the effect of menopause on the various hypothesis potentially explaining the occurrence of very-late onset schizophrenia in women.

Author Response

See attachment:

Round 2

Reviewer 2 Report

The authors have followed up all the suggestions and recommendations that were made. I consider that the paper can be published in its current form.